# Image-Based Phenotyping for Non-Destructive In Situ Rice (*Oryza sativa* L.) Tiller Counting Using Proximal Sensing

**DOI:** 10.3390/s22155547

**Published:** 2022-07-25

**Authors:** Yuki Yamagishi, Yoichiro Kato, Seishi Ninomiya, Wei Guo

**Affiliations:** Graduate School of Agricultural and Life Sciences, The University of Tokyo, Tokyo 113-8657, Japan; yamagishi-yuki008@g.ecc.u-tokyo.ac.jp (Y.Y.); snino@g.ecc.u-tokyo.ac.jp (S.N.)

**Keywords:** plant phenotyping, image analysis, synchronous emergence of leaf and tiller theory

## Abstract

The increase in the number of tillers of rice significantly affects grain yield. However, this is measured only by the manual counting of emerging tillers, where the most common method is to count by hand touching. This study develops an efficient, non-destructive method for estimating the number of tillers during the vegetative and reproductive stages under flooded conditions. Unlike popular deep-learning-based approaches requiring training data and computational resources, we propose a simple image-processing pipeline following the empirical principles of synchronously emerging leaves and tillers in rice morphogenesis. Field images were taken by an unmanned aerial vehicle at a very low flying height for UAV imaging—1.5 to 3 m above the rice canopy. Subsequently, the proposed image-processing pipeline was used, which includes binarization, skeletonization, and leaf-tip detection, to count the number of long-growing leaves. The tiller number was estimated from the number of long-growing leaves. The estimated tiller number in a 1.1 m × 1.1 m area is significantly correlated with the actual number of tillers, with 60% of hills having an error of less than ±3 tillers. This study demonstrates the potential of the proposed image-sensing-based tiller-counting method to help agronomists with efficient, non-destructive field phenotyping.

## 1. Introduction

Rice (*Oryza sativa* L.) is the staple food for nearly half of the population worldwide. Monitoring its yield is essential for regional and global food security, farmers’ livelihoods, and marketing strategies [1]. Advanced contracts with supply volume guarantees enable farmers to sell directly to restaurants and consumers at a high price. Additionally, farmers often adjust rice management practices, such as fertilizer applications to the crop, which necessitates real-time plant diagnosis.

Four main growth attributes for the grain yield of rice are (a) panicle number, (b) grain number per panicle, (c) ripening percentage, and (d) 1000-grain weight [2]. Tillers partially die, and the remainder become panicles. The number of tillers in the late growth stage is a key factor in determining panicle number. However, counting the number of rice panicles or tillers depends on manual work, which is labor-intensive. High throughput field-phenotyping techniques are expected to address such limitations.

In recent years, research on the image analysis of rice-panicle numbers has progressed remarkably (Table 1). In the early stages, the estimation of panicles from RGB images with feature-extraction methods was studied [3]. Their accuracy and availability have been rapidly improved from 50% to over 90% by applying deep-learning techniques, such as a Bayesian inference method, to eliminate the process of preparing supervisory data [4] and a density-map method instead of the object-detection method [5]. To date, 90% of the accuracy in rice-panicle counting can be expected using region-based fully convolutional networks (R-FCN) [6] and 95% by segmenting panicles, leaves, and background. However, it requires more sophisticated analysis than merely estimating panicle numbers [7].

In addition to panicle counting during rice harvesting, monitoring tiller numbers before flowering in yield estimation is imperative. Tiller counting at the vegetative and reproductive stages enables farmers to adjust crop management for rice growth [15]. However, only a few image analyses for rice-tiller counting have been conducted using pot-grown plants in a laboratory or greenhouse. There was an attempt to observe the structure of leaves and tillers in three dimensions by setting wheat plants on a turntable and collecting multi-angle images [16]. The tiller number was estimated by capturing images of the plant base [17]. For the image analysis in the field, some studies estimated the tiller number from rice-stubble images after harvesting [18,19]. No study has provided an effective tiller-counting method under flooded conditions, particularly around the panicle-formation stage. This is because of the difficulty in image processing due to the complete overlapping of neighboring plants. Counting tillers directly from the image is difficult because they are hidden behind the leaves.

Conversely, the emergence of lateral tillers in rice plants is highly synchronized with the leaf appearance on the main tiller [20,21]. Previous studies discovered that a new (lateral) tiller emerges from three nodes below the main tiller (Figure 1). Thus, when a new tiller emerges, the leaves that grew from the three upper nodes extend. That means the number of tillers should concomitantly increase with the number of leaves on the main tiller. The discovery of the empirical principle in tillering dynamics of rice was a milestone achievement in morphogenetic research of gramineous species [22], and various studies have confirmed this theory [23,24,25].

The ultimate goal is to develop a practical image-processing pipeline based on the empirical principle of rice morphogenesis. In particular, this study applies the basic rules of rice tillering to the algorithm in image analysis to estimate the tiller number in the field. The tiller number can be estimated by counting the long leaves of each hill, assuming that leaves growing from the main tiller reach a higher point and are visible at the top of the plant. The target period was set at the vegetative and reproductive stages when tillers that eventually bear panicles emerge [26]. When the rice is grown, it is difficult to observe tillers directly, so I observed leaves emerging from the main stem instead. This study is the first to apply biological knowledge of tillering behavior of rice plants into algorithm design, instead of only considering the image with complicated image-processing algorithms, such as deep learning.

## 2. Materials and Methods

### 2.1. Data Acquisition

Field experiments were conducted at the lowland farm of the Institute for Sustainable Agro-ecosystem Services, University of Tokyo, Tokyo, Japan (35°44′19.3” N 139°32′28.4” E). A popular japonica rice cultivar, Koshihikari, was grown in the summer of 2021. Twenty-one-day-old seedlings were transplanted on June 3 at a density of 21.2 hills m^−2^. Crop management, such as irrigation, fertilizer applications, and pest and disease control, followed Japan’s conventional rice farming systems [2]. The number of transplants per hill was arbitrarily varied from three to five seedlings to facilitate the evaluation of the effectiveness of the method. Images were acquired using an unmanned aerial vehicle (UAV), and the ground truth of the number of tillers on each hill was manually acquired (Figure 2). The UAV flew automatically and photographed the field with the onboard RGB camera. The right side of Figure 2 demonstrates the flight. Ground truth tiller numbers were observed for about 250 hills on each date in a 7 m × 6 m field plot.

For field survey and algorithm validation, poles with markers were set in the field to identify the location of hills in the field.

Aerial images were captured using a commercial-grade UAV (DJI Mavic Pro 2, Shenzhen, China) and an onboard camera with a resolution of 5472 × 3648 and a field of view of 77°. Poles with markers were set in the field to identify the location of hills. The images were captured on July 3 from a 1.5 m height with a 35° angle of elevation. The shooting angle was set as shallow as possible within the range where the rice in the foreground did not hide the rice base in the image. The shallower the angle, the more accurately the rice leaves could be measured automatically. Based on the height of the rice plants, the altitude of the flight, and the distance between the hills, a 35° angle of elevation is suitable for this experiment. Using the same method as that used for calculating the 35° angle of elevation, the angle of elevation was changed to 60° because the rice plants grew, and the plants in the front began to interfere with the plants in the back. The altitude was changed to 3 m, such that the downwash wind would not affect the plants in the shooting area at 60° elevation.

### 2.2. Analysis Method

Only the leaf tips above the momentum height were measured throughout the preliminary study (Figure 3). To calculate the momentum, each white pixel was weighted the same.
(1)$$N_{tiller+main\;stem}=N_{leaf\;on\;main\;stem}\approx N_{leaf\;tips\;above\;momentum\;height}$$

Considering the lens distortion and occlusion effect due to inadequate elevation angle, we only observed about 2 m^2^ (1 m × 2 m) in the center of a single image. Furthermore, seedlings that fluttered heavily in the wind were removed from the analysis. In the latter growth stage, rice plants tend to be affected by wind more, and more images were removed from the analysis. Finally, ten out of thirty patches were extracted and selected from each image.

A computer with a 64-bit Windows 10 operating system (Core i5 1.2 GHz, 8 GB RAM, Surface) was used to process and analyze the image data. The algorithm was written in Python using Google Colaboratory (Google Corporation, Menlo Park, CA, USA). The images acquired were saved in JPG file format. The flowchart of image processing is shown in Figure 4. These steps will be described further in the following sections. The following libraries were used for the thinning process (https://github.com/magikerwin1993/Line-Following-Python, accessed on 1 September 2021). The following libraries were used for the decorrelation stretching (https://github.com/lbrabec/decorrstretch, accessed on 10 September 2021). The rest of the program is self-coded. After ground truth tiller number and estimated leaf tip number were obtained, statistical analysis such as correlation coefficient analysis and *t* tests were conducted using Excel 2019 (Microsoft Corporation, Redmond, WA, USA).

Trimming and Thresholding: A field image was cropped into small patches with one hill. Only hills that could be seen without being considerably obscured by other hills were selected manually. Hills located from the bottom 25% to the middle of the vertical position in the field image were mostly used. Subsequently, a threshold-based method was utilized for green segmentation. The decorrelation stretching method was utilized to emphasize the color difference. Decorrelation stretching is a procedure for enhancing the color separation of an image with significant band-to-band correlation. The exaggerated colors easily improve visual interpretation. Binarization was applied based on the green value of each pixel. The same threshold value was used for each observation day. The threshold values were determined based on preliminary experiments and were set to 1.1–0.9 times the entire image average. Finally, a white pixel area with a size less than the threshold was removed. The threshold value was set to 500 pixels. All the leaves of a single plant were large, while some leaves with contour sizes of more than 100 pixels, reflected on the water surface, were falsely detected.

Skeletonization: This process was implemented using the line-following method. The two types of skeletonization methods are iterative and non-iterative. An iterative method is currently the most popular method; however, it often causes disordered boundary image processing problems. Non-iterative methods, which suit in-field images, are less sensitive to the smoothness of boundaries. The line-following approach draws a skeleton away from the leaf curve as the rice grows; however, it does not critically affect the detection of leaf tips in most cases.

Leaf-tip Counting: Leaf tips above the center of gravity were detected in the skeleton image. The pixels around the point were verified to determine if a pixel was a leaf tip. This is a method of scanning while setting up a window around the decision point. This study calls this the peak-in-window method (Figure 5). Subsequently, the hills average was calculated from the tiller number for each patch image. The pixel was adjudged a leaf tip when both of the following conditions were satisfied. The window sizes were fixed for all patch images through all observation dates in both Conditions 1 and 2.

Condition 1: There should be no skeleton in the window set above the judgment point because there should be no leaf area above a leaf tip (25 × 12 pixels above the judgment point).

In the image of this experiment, the width of the leaf appeared to be at least 12 pixels. The distance from the centerline of the leaf to the leaf edge would be at least six pixels. Therefore, two leaves that do not overlap should have a centerline distance of at least 6 + 6 pixels. The window size was set to 25 pixels in the horizontal direction to observe 12 pixels on each side.The window size in the horizontal direction was set to 12 pixels. The tip of the skeleton has a maximum distance of six pixels from the actual leaf tip, owing to the setting of the line-following method. Therefore, the distance between the skeleton tip of the non-overlapping leaf tip and the centerline of another leaf that interrupts the leaf tip is approximately 6 + 6 pixels.◦The line-following method sets a parameter for the interval between the placement of decision points. Each unevenness caused by a single shade is recognized as a tiller of the skeleton if the parameter value is significantly small. Conversely, the skeleton is drawn from multiple leaves if the set value is considerably large. For the images captured in this experiment, the best parameter setting in the preliminary study was seven pixels. When placing the decision point at a point seven pixels off, the tip of the skeleton will have a maximum of six-pixel separation from the actual leaf tip.

Condition 2: No point is recognized as a leaf edge in the window set around the judgment point. When the skeleton was zigzagged owing to the line-following method characteristic, multiple points on a single leaf were misclassified as leaf tips. To alleviate this error, a leaf tip with another leaf tip in the immediate neighborhood was eliminated from the leaf tip detection point (window size was set based on a preliminary experiment: horizontal direction was 31 pixels, and vertical direction was 20 pixels above the judgment point).

The flowchart of the peak-in-window method is described below (Figure 6).

## 3. Results

The results of image estimation of the number of tillers per hill were compared with those of the field observation of the manually measured value. Judging from the average of multiple hills, the standard deviation became smaller (Figure 7). For 25 hills’ average, the error variance dropped to about 1.0, and the error average dropped to 1.1 on 3 July and 7 August. The error average remained above 3 on 23 July.

The error of the proposed method was analyzed in two steps.

Step 1: Verify whether the true tiller number (A) can be estimated by manually counting leaf tips from RGB images (B).

Step 2: Compare the precision of manual leaf-tip counting (B) and automatic leaf-tip counting (C).

a.
**True tiller number (A)—Manually counted leaf-tip number (B)**


The effectiveness of the phenotyping approach was evaluated, which was inspired by the synchronous growth theory in the development of rice tillers (Figure 8). The tiller number measured manually in the field was compared with the leaf-tip number measured visually by a human from RGB images. The variance increased as the growth stage progressed; the number of visual measurements from the image was approximately 1 larger than the true tiller value, regardless of the growth stage. The pair-wise *t* test between true stem number and manual leaf-tip count implies that the difference is statistically significant (*p*-value showed less than 0.1%).

b.
**Manually counted leaf-tip number (B)**
**—Automatically counted leaf-tip number (C)**


The manually counted leaf-tip number from RGB images was compared with the automatically counted number. Similar to that in the automatic measurement, we calculated the momentum height, displayed the line on the image, and measured the leaf tip extending above it. The variance increased as the growth stage progressed; the automatic measurements were approximately 2–4 times larger than manual counting (Figure 9). The pair-wise *t* test between manual and automatic leaf-tip counts implies that the difference is statistically significant (*p*-value showed less than 0.1%).

c.
**True tiller number (A)—Automatically counted leaf-tip number (C)**


Finally, the error between the true tiller number and the automatically counted leaf-tip number was evaluated. The variance increased as the growth progressed; the automatic measurements were approximately 0.6–2.5 times larger than manual counting (Figure 10). The pair-wise *t* test between true tiller number and automatic leaf-tip count showed a *p*-value of 11% on 7 August, which implies that the difference is not statistically significant. On 7 and 23 July, the *t* test implies that the difference is statistically significant (*p*-value showed less than 0.001%).

The statistics of the observation results are described below (Table 2). For correlation coefficients, ground truth tiller numbers and manually counted leaf-tip numbers were 0.7–0.8 regardless of the growth stage; manual and automatically counted leaf-tip numbers were 0.5–0.3, and ground truth tiller number and automatic leaf-tip counting were 0.3–0.2. A *t* test between (A), (B), and (C) was also conducted, and the *p* value was less than 0.01 percent in most cases. There are statistically significant differences between the true tiller number (A), manually counted leaf-tip number (B), and automatically counted leaf-tip number (C).

## 4. Discussion

The tiller number can be estimated accurately from long leaf-tip counting using automated image analysis. For practical use, the accuracy is improved by estimating multiple plants per hill at once (Figure 7). Manual leaf-tip counting overestimates tiller numbers by evaluating the phenotyping approach (Figure 8). Additionally, automated counting underestimates compared to manual counting (Figure 9). The error bias of over- and undercounting offsets each other when comparing true tiller numbers with automatic leaf-tip counting. This results in moderate accuracy (Figure 10).

The error standard deviation became smaller with an increase in the hill number for average calculation; the error variance decreased to approximately one when 25 shares were observed. The error average did not decrease with an increase in the number of hills; it was approximately one on 3 July and 7 August, and three on 23 July (Figure 7). Tiller numbers per hill ranged from 5 to 17, with an average of 11 on August 7. Its range and average were nearly the same regardless of the day of observation. Since the tiller-number range was approximately 13, an error of 1–3 would not be a major practical problem. Considering the error of each hill, more than 60% of hills had an error of less than ±3 tillers regardless of the observation date (3 July: 81%, 23 July: 63%, 7 August: 66%).

The error factors were individually verified for each patch image in which the automated leaf-tip counting and tiller number deviated by four or more. Error causes could be categorized into four types (Figure 11):

Error (1) (overlapping leaves are seen as a single leaf) occurs most frequently and is considered the main cause of counting errors. It is difficult to distinguish the leaf tips o**f ov**erlapped leaves even by human judgment when observing the images after binarization. The approach of extracting the contour and boundary could achieve higher accuracy compared to the approach with binarization. Additionally, machine learning could be used to measure RGB images for higher accuracy.

Error (2) (invasion of neighboring hills’ leaves) does not explain the overcounting, but it decreases the correlation coefficient. Similar to Error (2), accuracy could be improved if the numbers of leaf tips of multiple plants were observed at once. Leaves reflected from neighboring plants based on the orientation of the leaves could be removed.

Error (3) (skeleton misfit and less peak count) occurs mainly at the left and right edges of the patch. The fitting of the skeleton fails when the leaves go out of the image. If we avoid cutting the image into pieces as patches and alter the approach to measuring multiple plants at once, higher accuracy can be achieved.

Error (4) (shooting elevation is sufficiently large that non-main stem leaves are counted): the long leaves growing from the main stem are a counting target, but they are also measured along with the short leaves together in some cases, which causes over counting. When observing rice with a large elevation angle, it is difficult to distinguish between long and short leaves. The drone’s flight altitude and position were not stable due to wind and caused large elevation angle shots.

A previous study found that the number of leaves on the main stem and the number of tillers have an error of about one leaf [25]. If the error mentioned above is reduced, more accurate observations can be made.

The wind conditions during UAV observation also seem vital in counting accuracy. For example, a high wind speed (3.2 m/s) on 23 July caused leaf disruption, which enlarged Errors (1) and (2). Furthermore, windy conditions complicate UAV altitude control, affecting the spatial resolution of images to be captured as well as image processing performance.

A versatile observation method could be implemented with less adjustment cost by utilizing crop science knowledge that has been tested for suitability in various cultivation environments. Although the data varied in terms of sunlight conditions and growth stage, it is easy to adjust the image-processing method. Only the binarization threshold parameter was adjusted for each observation date; the other parameters were consistently set to the same value.

The hill-trimming step was manually performed at the current stage. Several studies have reported the possibility of rice-hill extraction at early growth stages [27,28], but none for later stages when rice plants are heavily overlapped. We plan to develop such a whole-stage auto trimming algorithm to achieve a higher throughput of tillering counting.

As explained in the introduction, predicting the number of panicles from the tiller at an early stage is important. The data-assimilation method with a growth model is viable to improve the panicle-number prediction accuracy. Growth models such as WOFOST and ORYZA have been developed to simulate crop growth. WOFOST is a generic crop model that simulates many different crops using the same principles and algorithms, while ORYZA is a rice-specific model [29,30]. For example, WOFOST calculates attainable crop production for a location given knowledge about soil type, irradiation, temperature, plant characteristics, availability of water, and plant nutrients. Either model simulates major biological processes in order, which are phenological development, CO_2_-assimilation, respiration, and the partitioning of assimilates to the various organs. In this model, partitioning weight parameters between roots, stems, storage, and leaves are important parameters to describe crop conditions. Theoretically, the weight ratios of tillers are highly sensitive to prediction accuracy [31]. Tiller number counting could contribute to estimating current crop partitioning parameters by combining other observables such as biomass. The growth-model prediction could enhance accuracy through data assimilation with phenotyping results [32,33]. Data-assimilation methods with panicles and tillers could be further studied.

## 5. Conclusions

We proposed a simple image-processing pipeline that followed the empirical principles of synchronously emerging leaves and tillers in rice morphogenesis to non-destructively estimate the number of tillers at the vegetative and reproductive stages under flooded conditions. A non-destructive method for observing the tiller number of rice plants for this stage did not exist, and this method showed moderate accuracy. The estimated tiller number in a 1.1 m × 1.1 m area is significantly correlated with the actual number of tillers (error average = 1.02 of error standard deviation = 1.09 at the early reproductive stage), with 60% of hills having an error of less than ±3 tillers. As a novel research topic, the performance of the proposed method has been proven acceptable for practical usage and is expected to help agronomists with high throughput field-phenotyping tasks.

## Figures and Tables

**Figure 1 sensors-22-05547-f001:**
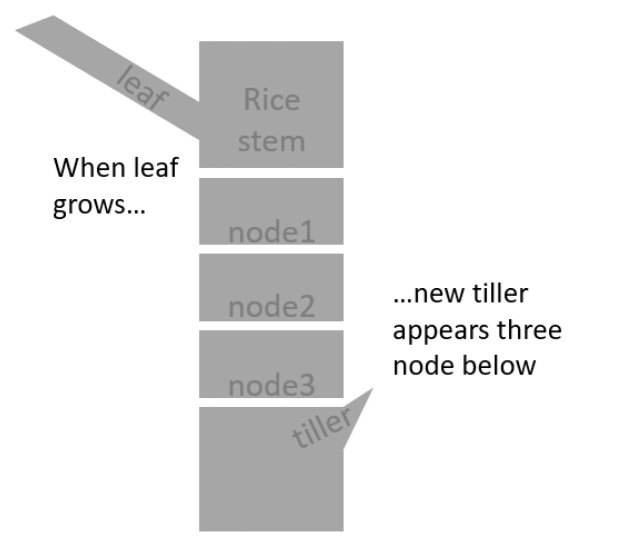
Synchronous growth theory in the development of rice tillers.

**Figure 2 sensors-22-05547-f002:**
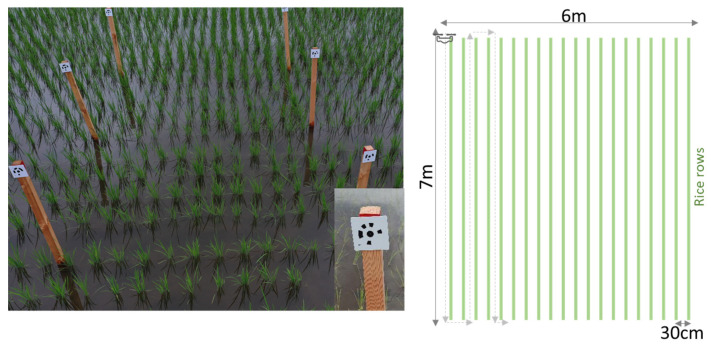
Experimental field. (Dotted arrows indicate flight paths).

**Figure 3 sensors-22-05547-f003:**
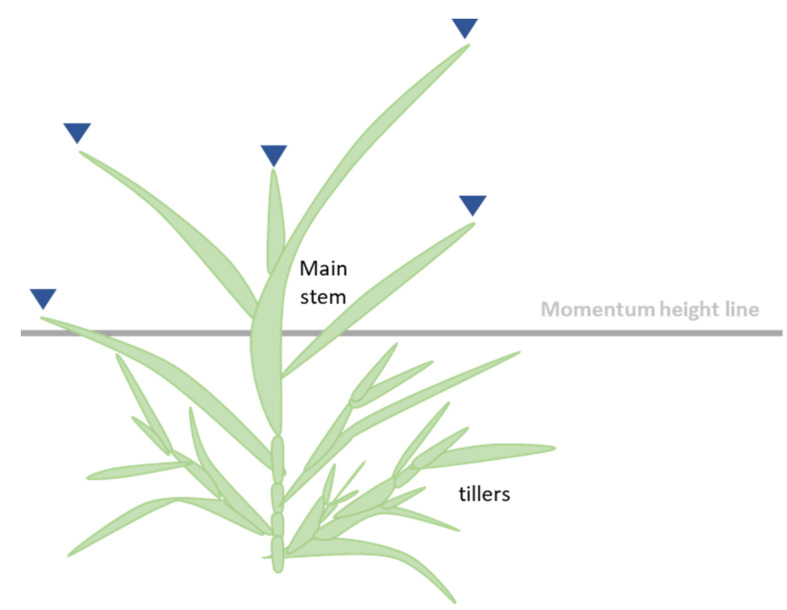
Phenotyping approach. Only the leaf tips above the momentum height (indicated by Inverted triangle) were counted.

**Figure 4 sensors-22-05547-f004:**
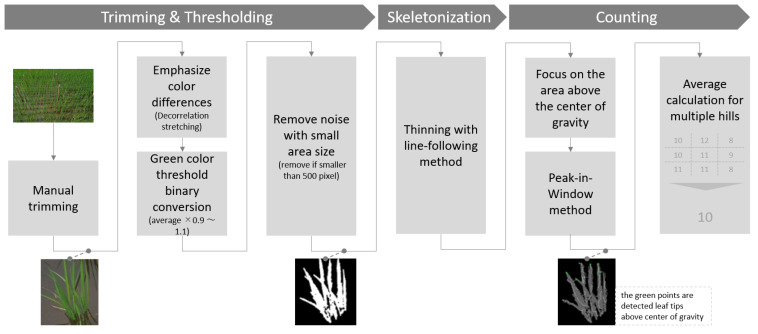
The overall process of image analysis.

**Figure 5 sensors-22-05547-f005:**
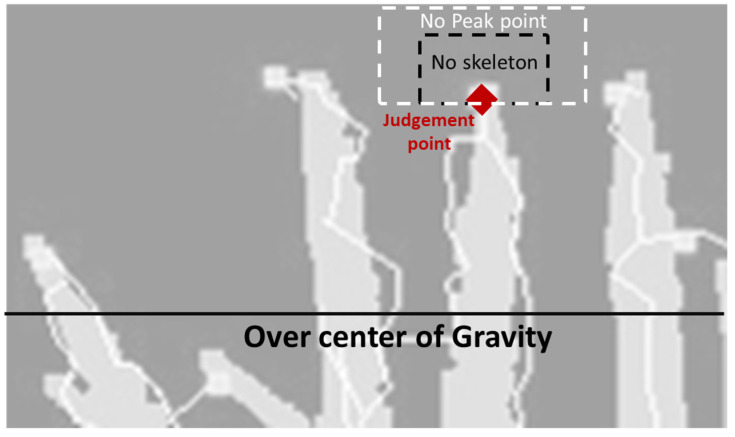
Concept of the peak-in-window method.

**Figure 6 sensors-22-05547-f006:**
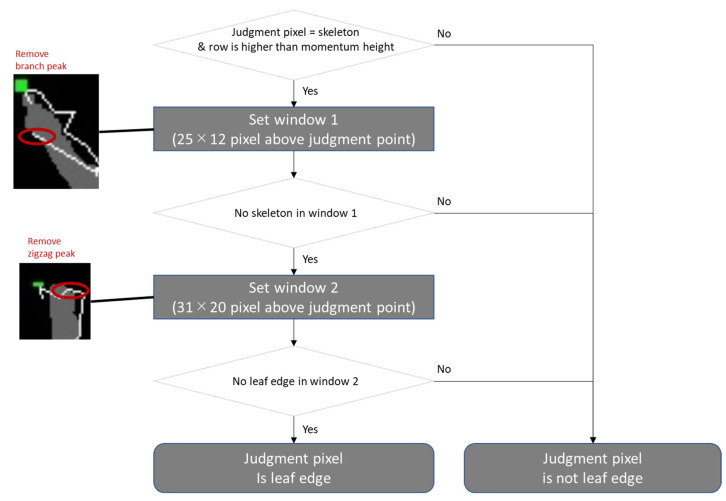
Flowchart of the peak-in-window method.

**Figure 7 sensors-22-05547-f007:**
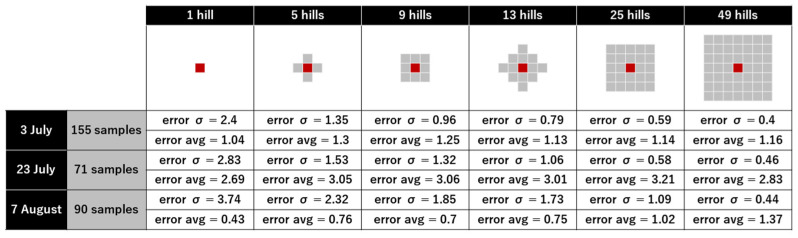
Accuracy of tiller-number estimation of multiple hills’ average (When there are various estimates for the same hill, the average value was used. Only observable hills in the range were considered. Data with observed values in more than one-third of the range were deemed to be valid. Observed average of grayed-out hills around red-painted hills.).

**Figure 8 sensors-22-05547-f008:**
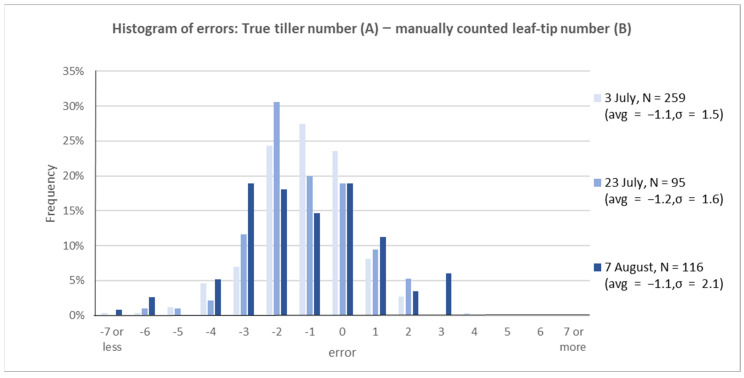
Histogram of errors between true tiller number and manually counted leaf-tip number counting (even if they were the same hill, different images were analyzed as another sample. The sample number is not consistent with that in Figure 7).

**Figure 9 sensors-22-05547-f009:**
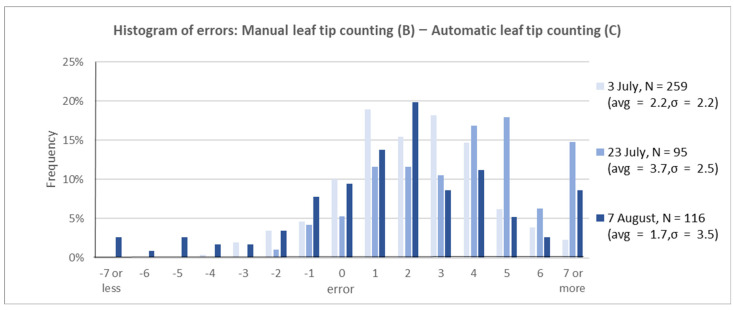
Error histogram between manually and automatically counted leaf-tip numbers.

**Figure 10 sensors-22-05547-f010:**
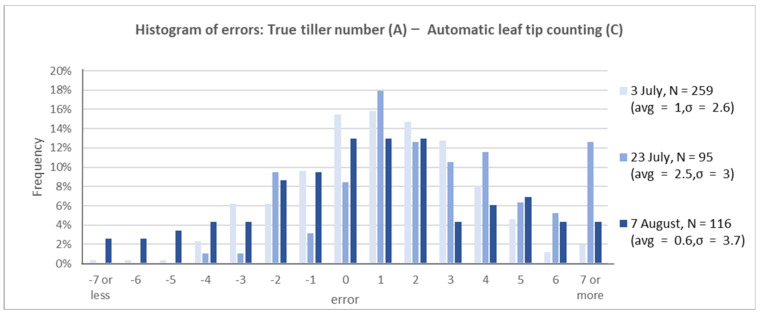
Error histogram between true tiller number and automatically counted leaf-tip number.

**Figure 11 sensors-22-05547-f011:**
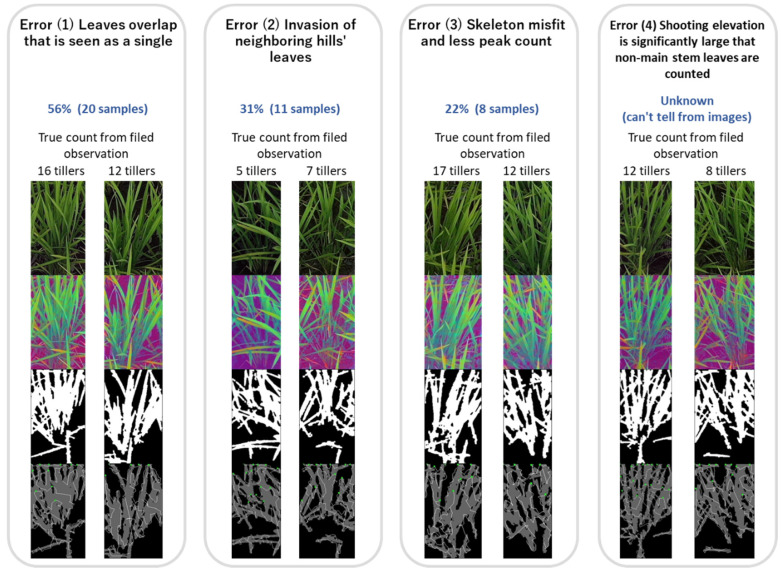
Error type in automatic tiller counting. Undercounting due to leaf-region overlap, overcounting due to invasion of neighboring hills’ leaves, undercounting due to skeleton misfit, and overcounting due to large shooting elevation angles, which counts non-main stem leaves (error larger than four, August 7 observation). In bottom image, white lines represent skeleton of leaves, and green dots represent points which is regarded as leaf tips above center of gravity line.

**Table 1 sensors-22-05547-t001:** Summary of various methods and their accuracy for **panicle** counting (not tiller counting).

Accuracy		Method		Reference
Recall	Precision	Imaging	Analysis	
50%	Not Available	fixed-point camera	SVM	[3]
88%	87%	UAV	improved R-FCN	[6]
95%	75%	UAV	Unsupervised Bayesian learning	[4]
82.50%	Not Available	field scanner	SVM	[8]
95%	56%	fixed-point camera	SVM	[9]
94%	Not Available	turntable camera	ANN	[10]
73%	82%	GAV	SVM	[11]
89%	82%	fixed-point camera	CNN	[12]
70%	85%	fixed-point camera	CNN	[13]
62%	99.6%	fixed-point camera	CNN	[14]

**Table 2 sensors-22-05547-t002:** Detailed table of observation results.

		True Tiller Number—Manual Leaf Tip Counting			Manual Leaf Tip Counting—Automatic Leaf Tip Counting			True Tiller Number—Automatic Leaf Tip Counting		
		3 July	23 July	7 August	3 July	23 July	7 August	3 July	23 July	7 August
R (correlation coefficient)		0.73	0.83	0.73	0.49	0.47	0.30	0.27	0.25	0.21
error average		−1.12	−1.16	−1.14	2.15	3.68	1.70	1.03	2.53	0.56
error variance		1.53	1.55	2.06	2.18	2.52	3.46	2.55	2.99	3.68
*t*-test *p* value		0.00	0.00	0.00	0.00	0.00	0.00	0.00	0.00	0.11
Histogram of errors (sample number for each error)	−7 or less	1	0	1	0	0	3	1	0	3
	−6	1	1	3	0	0	1	1	0	3
	−5	3	1	0	0	0	3	1	0	4
	−4	12	2	6	1	0	2	6	1	5
	−3	18	11	22	5	0	2	16	1	5
	−2	63	29	21	9	1	4	16	9	10
	−1	71	19	17	12	4	9	25	3	11
	0	61	18	22	26	5	11	40	8	15
	1	21	9	13	49	11	16	41	17	15
	2	7	5	4	40	11	23	38	12	15
	3	0	0	7	47	10	10	33	10	5
	4	1	0	0	38	16	13	21	11	7
	5	0	0	0	16	17	6	12	6	8
	6	0	0	0	10	6	3	3	5	5
	7 or more	0	0	0	6	14	10	5	12	5

## Data Availability

Not applicable.

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
