# Peer review of "Image-Based Phenotyping for Non-Destructive In Situ Rice (Oryza sativa L.) Tiller Counting Using Proximal Sensing"

_sensors, 2022, doi:10.3390/s22155547_

Round 1

Reviewer 1 Report

Image-based phenotyping is one of novel and newly developed study domains that has broad applications in agriculture. The obtained results of the study are quite convincing. However, research methodology and results section need to be improved significantly by providing additional materials. Section 4 - Discussion states that WOFOST and ORYZA models were developed, but these models are not given. It is necessary to provide these models and elaborate them respectively. Moreover, in Results, some cross correlation analysis of the obtained data of tiller number and leaf tip number by a manual count and an automatic count.

Another crucial point that requires a special attention is that error margins of the proposed simple imaging technique are much higher than other benchmarked methods. The authors should explain the key advantages of their proposed methodology over other machine learning based methods which have much higher accuracy. 

Conclusions Section needs to be improved after updating the whole manuscript.  

Author Response

Dear editor and Reviewers, thank you for your constructive comments. We have now improved our manuscript based on your suggestions, and please find the response in detail below:

I am sorry that the image file may not have been inserted properly, and would appreciate it if you could check it with the attached word file.

=============================================

Comment: Image-based phenotyping is one of novel and newly developed study domains that has broad applications in agriculture. The obtained results of the study are quite convincing. However, research methodology and results section need to be improved significantly by providing additional materials. In Results, some cross correlation analysis of the obtained data of tiller number and leaf tip number by a manual count and an automatic count.

Response : Thank you for your comments.

For materials and methods, three materials are added to elaborate the detail of the manuscript.

    • Experimental design were added to explain flight path and settings. Underlined statement is the description added from previous version.

Field experiments were conducted at the lowland farm of the Institute for Sustainable Agro-ecosystem Services, University of Tokyo, Tokyo, Japan (35°44’19.3”N 139°32’28.4”E). A popular japonica rice cultivar, Koshihikari, was grown in the summer of 2021. Twenty-one-day-old seedlings were transplanted on June 3 at a density of 21.2 hills m-2. Crop management, such as irrigation, fertilizer applications, pest, and disease control, followed Japan's conventional rice farming systems [2]. The number of transplants per hill was arbitrarily varied from three to five seedlings to facilitate the evaluation of the effectiveness of the method. Images were acquired using an unmanned aerial vehicle (UAV), and the ground truth of the number of tillers on each hill was manually acquired (Figure 2). The UAV flew automatically and photographed the field with the onboard RGB camera. The waypoint of the flight was set up to fly perpendicular to the rows. right side of figure 2 demonstrates the flight. Ground truth tiller numbers were observed for about 250 hills on each date in a 7m x 6m field plot.

 (please look at attached file to see figure.)

Figure 2. Experimental field.

    • Selection of patch image is elaborated in the following descriptions.

Furthermore, seedlings that fluttered heavily in the wind were removed from the analysis. In the latter growth stage, rice plants tend to be affected by wind heavier, and more images were removed from the analysis. Finally,  ten out of thirty patches were extracted and selected from each image.

    • Statistical method and replication, software tools were elaborated in the following descriptions. 

A computer with a 64-bit Windows 10 operating system (Core i5 1.2 GHz, 8 GB RAM, Surface) was used to process and analyze the image data. The algorithm was written in Python using Google Colaboratory (Google Corporation, Menlo Park, CA, USA). The images acquired were saved in a JPG file format. The flowchart of image processing is shown in (Figure 4). These steps will be described further in the following sections. The following libraries were used for the thinning process. (https://github.com/magikerwin1993/Line-Following-Python) The following libraries were used for the decorrelation stretching. (https://github.com/lbrabec/decorrstretch) The rest of the program is self-coded. After ground truth tiller number and estimated leaf tip number were obtained, statistical analysis such as correlation coefficient analysis and T-tests were conducted using Excel 2019 (Microsoft Corporation, Redmond, WA, USA).

    • In result section, statistical analysis is elaborated. Underlined description is added.

The statistics of the observation results are described below (Table 2). For correlation coefficients, ground truth tiller number and manually counted leaf-tip number were 0.7–0.8 regardless of the growth stage; manual and automatically counted leaf-tip number were 0.5–0.3, and ground truth tiller number and automatic leaf-tip counting was 0.3–0.2. T-test between (A) (B) (C) was also conducted and the P value was less than 0.01 percent in most cases. True tiller number (A), Manually counted leaf-tip number (B), Automatically counted leaf-tip number (C) has statistically significant differences.

Comment: Section 4 - Discussion states that WOFOST and ORYZA models were developed, but these models are not given. It is necessary to provide these models and elaborate them respectively.

Response : Thank you for your suggestion. WOFOST and ORYZA are elaborated in underlined descriptions below. Either model simulates major biological processes such as phenological development, CO2-assimilation, leaf dynamics, transpiration, respiration, partitioning of assimilates to the various organs, and dry matter formation. ORYZA omits detail simulation for some processes because it is a rice-specialized model. Schematics of the two models are shown below, and those two simulations’ structure are basically same, but WOFOST is applicable for many crops, while ORYZA is for rice only.

As explained in the introduction, predicting the number of panicles from the tiller at the early stage is important. The data-assimilation method with a growth model is viable to improve the panicle-number prediction accuracy. Growth models such as WOFOST and ORYZA have been developed to simulate crop growth. WOFOST is a generic crop model which simulates many different crops using the same principles and algorithms, while ORYZA is a rice-specific model [29, 30]. For example, WOFOST calculates attainable crop production for a location given knowledge about soil type, irradiation, temperature, plant characteristics, availability of water, and plant nutrients. Either model simulates major biological processes in order, which are phenological development, CO2-assimilation, respiration, partitioning of assimilates to the various organs. In this model, partitioning weight parameters between roots, stems, storage, and leaves are important parameters to describe crop conditions. Theoretically, the weight ratios of tillers are highly sensitive to prediction accuracy [31]. Tiller number counting could contribute to estimating current crop partitioning parameters by combining other observables such as biomass. The growth-model prediction could enhance accuracy through data assimilation with phenotyping results [32, 33]. Data-assimilation methods with panicles and tillers could be further studied.

 (please look at attached file to see figure.)

WOFOST(https://www.wur.nl/en/show/A-gentle-introduction-to-WOFOST.htm)

 (please look at attached file to see figure.)

ORYZA(https://www.researchgate.net/publication/40199767_ORYZA-W_Rice_growth_model_for_irrigated_and_rainfed_environments)

Comment: Another crucial point that requires a special attention is that error margins of the proposed simple imaging technique are much higher than . The authors should explain the key advantages of their proposed methodology over other machine learning based methods which have much higher accuracy. Conclusions Section needs to be improved after updating the whole manuscript. 

Response : The advantage of the proposed method is that the tiller number can be estimated through the vegetative and reproductive period, including growth stage when the entire plant is not visible. Also, the proposed method shows in-situ observation without destructing seedlings. This novelty is additionally explained in the following section with underlined description.

We proposed a simple image-processing pipeline that followed the empirical principles of synchronously emerging leaves and tillers in rice morphogenesis to non-destructively estimate the number of tillers at the vegetative and reproductive stages under flooded conditions. A nondestructive in-situ method for observing the tiller number of rice plants through vegetative and reproductive periods is presented, which is the novelty of this research.

To achieve this goal, long leaf tips were observed instead of observing tillers directly, which is a novelty of this research. This approach is inspired by synchronous growth theory, which is explained in detail in following part.

The ultimate goal is to develop a practical image-processing pipeline based on the empirical principle of rice morphogenesis. Particularly, this study applies the basic rules of rice tillering to the algorithm in image analysis to estimate the tiller number in the field. The tiller number can be estimated by counting the long leaves of each hill assuming that leaves growing from the main tiller reach a higher point and are visible at the top of the plant. The target period was set at the vegetative and reproductive stages when tillers that eventually bear panicles emerge [26]. When the rice is grown, it is difficult to observe tillers directly, so I observed leaves emerging from the main stem instead. This study is the first to apply biological knowledge of tillering behavior of rice plants into algorithm design, instead of only considering the image with complicated image-processing algorithms, such as deep learning.

 (please look at attached file to see figure.)

Figure 1: Synchronous growth theory in the development of rice tillers.

Also,  Table 1 shows the accuracy of panicle counting, not tiller counting, which is very confusing. The bold letter is added to reduce misunderstandings.

Table 1. Summary of various methods and its accuracy for panicle counting (not tiller counting).

Accuracy

Method

Reference

Recall

Precision

Imaging

Analysis

50%

NA

fixed-point camera

SVM

[3]

88%

87%

UAV

improved R-FCN

[6]

95%

75%

UAV

Unsupervised Bayesian learning

[4]

82.50%

NA

field scanner

SVM

[8]

95%

56%

fixed-point camera

SVM

[9]

94%

NA

turntable camera

ANN

[10]

73%

82%

GAV

SVM

[11]

89%

82%

fixed-point camera

CNN

[12]

70%

85%

fixed-point camera

CNN

[13]

62%

99.6%

fixed-point camera

CNN

[14]

We appreciate your review.

Reviewer 2 Report

Overall Comments

Define all acronyms the first time they are introduced.

Apply the SI system writing rules, leaving a space between value and unit

Materials and methods section required inclusion of explanation of detailed experimental design, about statistical method and treatment replication, field size and crop variety.

Specific Comments

L2: In the title, in-situ word might be more appropriate rather than in-field. Suggestion to change the title if possible.

L7:   remove (branches). Word tillers itself explanatory for the scientific community.

L8: However, this is measured only by manual counting of emerging tillers. This sentence requires rewriting. Add how drudgery and time-consuming process of manual counting.

L13: First, we fly an unmanned aerial vehicle. Rewrite this sentence. Like An unmanned aerial vehicle was used to capture the field images at a flying height of 1.5- 3 m above the rice canopy.   Second, this flying height 1.5- 3 m above the ground is a very low flying height for UAV imaging. Kindly Check it. Also clarify are you taking levels between flying heights. I think is two height so write 1.5 and 3.0 m.

L14: Don’t use We, first, finally words. Kindly check through of manuscript.

L16: Write or start a sentence like Tiller number was estimated from…………

L17: (a) (b) (c) and (d) remove. Write continuously with , .

L46: Table 1 caption needs rewriting. Write “Summary of various methods and its accuracy for tiller counting”. Also, Shift column 1 viz. Paper to last. Change name Papers to Reference or source. Change table font size as per manuscript guidelines.

L52: Remove the author name that is Fang et al. Rewrite the sentence.

L54: Same remove the author name and rewrite. Put reference [17] in last.          

L56: So L56 to L59 So why you are not facing a problem like this. What unique method you apply in this study.

L86: Is it necessary to write in italic “japonica”?

L87: Three to five seedlings were transplanted? Kindly check or rewrite the sentence. Thee to five seedlings between poles? Distance between seedlings?

L88: m-2 write this way. Use SI unit throughout the manuscript.

L89: followed the standard practices in Japan. If possible, add reference related to published documents of practices for rice.

L91, L98, L99: Mention Flightpath, autonomous flight, or manual flight, Which camera was used (RGB, Multispectral??). Write FOV full form.

L105: Based on the height of the rice plants, altitude of 104 the flight, and distance between the hills, we discovered that a 35° angle of elevation is 105 suitable for this experiment.  Have you conducted experiments for standardization of 30o angle of elevation? If yes write brief about method. 2-3 sentences.

L115: is it an equation? If yes add Eqn numbering

L111: Add some methodology in the Phenotyping sub-section.

L119: Elaborate. Add how image analyisis was conducted. What tools or softwere required for image analysis. How many iamges required for single desired output? Add these type of information.

L121: If possible increase font size in figure. Difficult to read.

L123: In one field image how much area was captured?

L127 to 135: Mention which Software used for threshold-based method or Decorrelation stretching and  Binarization? Similar way for Skeletonization  and Leaf-tip Counting.

L152: 25 × 12 pixels these values or window size were fix for all images? Or might be change in if we use different image. Clarify please.

L182: write this sentence above Figure 5 in continuation of Condition 2. (after Condition 2)

L186: Materials and Methods requires improvements. Add detailed field experimental design. Overall schemetic view of field. No mention of statistical method, Replications.

L191: Elaborate the Results portion. Write in continuous.

…………..observation of the manually measured value. Judging from the average of multiple…………………….

 L194: Why samples were different size? 155, 71 and 90? Why not same for three date?

L210: Yes, please mention statistical method and softwere used in materials and methods section.

L214: Increase Fig. 8 size. If possible, make the axis in black color for more visibility. Same for Fig. 9 and 10 and further.

L218: Instade of this (B) Manually counted leaf-tip number – (C)Automatically counted leaf-tip number

Write comparison between Manually counted leaf-tip number(B)  and Automatically counted leaf-tip number(C).

L204: Denote (A) (B) (C) and (D) at the end. Like True tiller number (A)

L299: increase font size in Fig. 11

L329: Improve Conclusion. Add data brief data about the experiments. Tiller count and error data.

L349: Kindly check references style with journal guideline.

Author Response

Dear editor and Reviewers, thank you for your constructive comments. We have now improved our manuscript based on your suggestions, and please find the response in detail below:

===============================

Define all acronyms the first time they are introduced.

Response : We apologize for unclear expression. R-FCN, UAV, FOV are defined or written in full form. Please check underlined statement below.

Images were acquired using an unmanned aerial vehicle (UAV)

To date, 90% of the accuracy in rice-panicle counting can be expected using Region-based Fully Convolutional Networks (R-FCN)

Aerial images were captured using a commercial-grade UAV (DJI Mavic Pro 2, Shenzhen, China) and an onboard camera with a resolution of 5472 × 3648 and a Field of View 77°.

Apply the SI system writing rules, leaving a space between value and unit

Response : The description has been modified. In case of elevation angle and FOV of the camera, I would like to use ° instead of rad, following other papers format.

Materials and methods section required inclusion of explanation of detailed experimental design, about statistical method and treatment replication, field size and crop variety.

Response : Overall schematic view of the field was added in Figure2. Also L96 and L126 are added to explain the detail of the experiment.

Field experiments were conducted at the lowland farm of the Institute for Sustainable Agro-ecosystem Services, University of Tokyo, Tokyo, Japan (35°44’19.3”N 139°32’28.4”E). A popular japonica rice cultivar, Koshihikari, was grown in the summer of 2021. Twenty-one-day-old seedlings were transplanted on June 3 at a density of 21.2 hills m-2. Crop management, such as irrigation, fertilizer applications, pest, and disease control, followed Japan's conventional rice farming systems [2]. The number of transplants per hill was arbitrarily varied from three to five seedlings to facilitate the evaluation of the effectiveness of the method. Images were acquired using an unmanned aerial vehicle (UAV), and the ground truth of the number of tillers on each hill was manually acquired (Figure 2). The UAV flew automatically and photographed the field with the onboard RGB camera. The waypoint of the flight was set up to fly perpendicular to the rows. right side of figure 2 demonstrates the flight. Ground truth tiller numbers were observed for about 250 hills on each date in a 7m x 6m field plot.

Considering the lens distortion and occlusion effect due to inadequate elevation angle,  we only observed about 2m2 (1m × 2m) area in the center of a single image. Furthermore, seedlings that fluttered heavily in the wind were removed from the analysis. In the latter growth stage, rice plants tend to be affected by wind heavier, and more images were removed from the analysis. Finally,  ten out of thirty patches were extracted and selected from each image.

A computer with a 64-bit Windows 10 operating system (Core i5 1.2 GHz, 8 GB RAM, Surface) was used to process and analyze the image data. The algorithm was written in Python using Google Colaboratory (Google Corporation, Menlo Park, CA, USA). The images acquired were saved in a JPG file format. The flowchart of image processing is shown in (Figure 4). These steps will be described further in the following sections. The following libraries were used for the thinning process. (https://github.com/magikerwin1993/Line-Following-Python) The following libraries were used for the decorrelation stretching. (https://github.com/lbrabec/decorrstretch) The rest of the program is self-coded. After ground truth tiller number and estimated leaf tip number were obtained, statistical analysis such as correlation coefficient analysis and T-tests were conducted using Excel 2019 (Microsoft Corporation, Redmond, WA, USA).

(please look at the attachment word file to see image)

L2: In the title, in-situ word might be more appropriate rather than in-field. Suggestion to change the title if possible.

Response : We appreciate the feedback and decided to use “in-situ” instead. The title has now been changed to “Image-based phenotyping for nondestructive in-situ rice (Oryza Sativa L.) tiller counting using proximal sensing” from “Image-based phenotyping for nondestructive in-field rice (Oryza Sativa L.) tiller counting using proximal sensing”.

L7:   remove (branches). Word tillers itself explanatory for the scientific community.

Response : The description removed in the sentence below.

Abstract: The increase in the number of tillers (branches) of rice significantly affects grain yield.

Each unevenness caused by a single shade is recognized as a tiller branch of the skeleton if the parameter value is significantly small

L8: However, this is measured only by manual counting of emerging tillers. This sentence requires rewriting. Add how drudgery and time-consuming process of manual counting.

Response : The passage has been modified for better clarity. “, where the most common method is to count by touching with the hand”

The increase in the number of tillers of rice significantly affects grain yield. However, this is measured only by manual counting of emerging tillers, where the most common method is to count by hand touching method.

L13: First, we fly an unmanned aerial vehicle. Rewrite this sentence. Like An unmanned aerial vehicle was used to capture the field images at a flying height of 1.5- 3 m above the rice canopy.   Second, this flying height 1.5- 3 m above the ground is a very low flying height for UAV imaging. Kindly Check it. Also clarify are you taking levels between flying heights. I think is two height so write 1.5 and 3.0 m.

Response : I appreciate your suggestion. The description is modified as “An unmanned aerial vehicle was used to capture the field images at a flying height of 1.5 or 3 m above the rice canopy, which is a very low height for UAV imaging”, to explain its height more precisely and to tell its height is very low.

L14: Don’t use We, first, finally words. Kindly check through of manuscript.

L16: Write or start a sentence like Tiller number was estimated from…………

Response : Thank you for your valuable suggestions. We have now improved the writing such as.

“We fly an unmanned aerial vehicle 1.5-3 m above the rice canopy to capture the images.”→ An unmanned aerial vehicle was used to capture the field images at a flying height of 1.5 or 3 m above the rice canopy, which is a very low height for UAV imaging

“Finally, we estimate the tiller number from the number of long-growing leaves.” The tiller number was estimated from the number of long-growing leaves.

L17: (a) (b) (c) and (d) remove. Write continuously with , .

Response : Thank you for your suggestion. “binarization, skeletonization, and leaf-tip detection”  is modified into “binarization, skeletonization, leaf-tip detection”

L46: Table 1 caption needs rewriting. Write “Summary of various methods and its accuracy for tiller counting”. Also, Shift column 1 viz. Paper to last. Change name Papers to Reference or source. Change table font size as per manuscript guidelines.

Response : I appreciate your suggestion. I have changed the column order and table format. Also changed caption. Please note that this table shows panicle observation accuracy but not tiller counting accuracy. There are only few research for tiller observation. Please look at the underlined description in Table 1 in the paper.

Table 1. Summary of various methods and its accuracy for panicle counting (not tiller counting).

Accuracy

Method

Reference

Recall

Precision

Imaging

Analysis

50%

NA

fixed-point camera

SVM

[3]

88%

87%

UAV

improved R-FCN

[6]

95%

75%

UAV

Unsupervised Bayesian learning

[4]

82.50%

NA

field scanner

SVM

[8]

95%

56%

fixed-point camera

SVM

[9]

94%

NA

turntable camera

ANN

[10]

73%

82%

GAV

SVM

[11]

89%

82%

fixed-point camera

CNN

[12]

70%

85%

fixed-point camera

CNN

[13]

62%

99.6%

fixed-point camera

CNN

[14]

L52: Remove the author name that is Fang et al. Rewrite the sentence.

L54: Same remove the author name and rewrite. Put reference [17] in last.          

Response : I appreciate your suggestion, and I followed the guidance above.

There was an attempt to observe the structure of leaves and tillers in three dimensions by setting wheat plants on a turntable and collecting multi-angle images[16]. The tiller number was estimated by capturing images of the plant base in [17].”

L56: So L56 to L59 So why you are not facing a problem like this. What unique method you apply in this study.

Response : My approach is to count leaf tips on the main stem, not to count tillers directly. When the rice is grown, it is difficult to observe tillers, so I observed leaves emerging from the stem instead (Figure 3 illustrates my approach). This approach is inspired by the synchronous growth theory provided by Katayama(Katayama, T. [Studies on Rice and Wheat Tillers]. Ine Mugi No Bungetsu Kenkyu (in Japanese). Yokendo. 1951.). The novelty of my study is to discover effective approach to observe tillers when rice is grown. The following description was added in the last part of the introduction section at the line of 77.

When the rice is grown, it is difficult to observe tillers directly, so I observed leaves emerging from the main stem instead.

L86: Is it necessary to write in italic “japonica”?

Response : I apologize for my miss description. It is customary to use the Italic format for crop variety, but Japonica is not a variety, so I modified it in a non-italic description.

L87: Three to five seedlings were transplanted? Kindly check or rewrite the sentence. Thee to five seedlings between poles? Distance between seedlings?

Response : I apologize for my unclear description. Three to five is the number of seedlings per hill. Description was deleted and moved to another sentence at line 93:

The number of transplants per hill was arbitrarily varied from three to five seedlings to facilitate the evaluation of the effectiveness of the method.

L88: m-2 write this way. Use SI unit throughout the manuscript.

Response : The description has been modified. In case of elevation angle and FOV of camera, I would like to use ° instead of rad, following other paper format.

L89: followed the standard practices in Japan. If possible, add reference related to published documents of practices for rice.

Response : Cited [2] as a reference describing common rice cultivation methods.

L91, L98, L99: Mention Flightpath, autonomous flight, or manual flight, Which camera was used (RGB, Multispectral??). Write FOV full form.

Response : Thank you for your suggestion. The following description were added.

The UAV flew automatically and photographed the field with the onboard RGB camera. The waypoint of the flight was set up to fly perpendicular to the rows. right side of figure 2 demonstrates the flight. Ground truth tiller numbers were observed for about 250 hills on each date in a 7m x 6m field plot.

L105: Based on the height of the rice plants, altitude of 104 the flight, and distance between the hills, we discovered that a 35° angle of elevation is 105 suitable for this experiment.  Have you conducted experiments for standardization of 30angle of elevation? If yes write brief about method. 2-3 sentences.

Response : I appreciate your kind suggestion. The following statement were added.

The UAV flew automatically and photographed the field with the onboard RGB camera. The waypoint of the flight was set up to fly perpendicular to the rows. right side of figure 2 demonstrates the flight. Ground truth tiller numbers were observed for about 250 hills on each date in a 7m x 6m field plot.

L115: is it an equation? If yes add Eqn numbering

Response : Yes, it is an equation. Number is added to the equation.

L111: Add some methodology in the Phenotyping sub-section.

Response : Phenotyping sub-section is now integrated and unified into the Analysis method subsection.

L119: Elaborate. Add how image analysis was conducted. What tools or software required for image analysis. How many images required for single desired output? Add these type of information.

Response : I appreciate your kind suggestion. Following description was added.

A computer with a 64-bit Windows 10 operating system (Core i5 1.2 GHz, 8 GB RAM, Surface) was used to process and analyze the image data. The algorithm was written in Python using Google Colaboratory (Google Corporation, Menlo Park, CA, USA). The images acquired were saved in a JPG file format. The flowchart of image processing is shown in (Figure 4). These steps will be described further in the following sections. The following libraries were used for the thinning process. (https://github.com/magikerwin1993/Line-Following-Python) The following libraries were used for the decorrelation stretching. (https://github.com/lbrabec/decorrstretch) The rest of the program is self-coded. After ground truth tiller number and estimated leaf tip number were obtained, statistical analysis such as correlation coefficient analysis and T-tests were conducted using Excel 2019 (Microsoft Corporation, Redmond, WA, USA).

L121: If possible increase font size in figure. Difficult to read.

Response : Font size is enlarged now as below.

(please look at the attachment word file to see image)

L123: In one field image how much area was captured?

Response : About 6.3m2 was observed from a single image. The following description has been added to the analysis method.

Considering the lens distortion and occlusion effect due to inadequate elevation angle,  we only observed about 2m2 (1m × 2m) area in the center of a single image. Furthermore, seedlings that fluttered heavily in the wind were removed from the analysis. In the latter growth stage, rice plants tend to be affected by wind heavier, and more images were removed from the analysis. Finally,  ten out of thirty patches were extracted and selected from each image.

L127 to 135: Mention which Software used for threshold-based method or Decorrelation stretching and  Binarization? Similar way for Skeletonization  and Leaf-tip Counting.

Response : Libraries are used for Decorrelation stretching and Skeletonization. Binarization and Leaf-tip counting were conducted through self coded program written in Python using Google Colaboratory. Please refer detailed information in the following description.

A computer with a 64-bit Windows 10 operating system (Core i5 1.2 GHz, 8 GB RAM, Surface) was used to process and analyze the image data. The algorithm was written in Python using Google Colaboratory (Google Corporation, Menlo Park, CA, USA). The images acquired were saved in a JPG file format. The flowchart of image processing is shown in (Figure 4). These steps will be described further in the following sections. The following libraries were used for the thinning process. (https://github.com/magikerwin1993/Line-Following-Python) The following libraries were used for the decorrelation stretching. (https://github.com/lbrabec/decorrstretch) The rest of the program is self-coded. After ground truth tiller number and estimated leaf tip number were obtained, statistical analysis such as correlation coefficient analysis and T-tests were conducted using Excel 2019 (Microsoft Corporation, Redmond, WA, USA).

L152: 25 × 12 pixels these values or window size were fix for all images? Or might be change in if we use different image. Clarify please.

Response : The window size were fixed for all date and all patch image. The following description was added to clarify this point.

The pixel was adjudged a leaf tip when both of the following conditions were satisfied. The window sizes were fixed for all patch images through all observation dates in both Conditions 1 and 2.

L182: write this sentence above Figure 5 in continuation of Condition 2. (after Condition 2)

Response : The sentence is moved to the place mentioned above.

L186: Materials and Methods requires improvements. Add detailed field experimental design. Overall schemetic view of field. No mention of statistical method, Replications.

Response : Following descriptions were added to elaborate field experimental design and statistical method and replications.

  1. Experimental design were added to explain flight path and UAV settings. Underlined statement is the description added from previous version.

Field experiments were conducted at the lowland farm of the Institute for Sustainable Agro-ecosystem Services, University of Tokyo, Tokyo, Japan (35°44’19.3”N 139°32’28.4”E). A popular japonica rice cultivar, Koshihikari, was grown in the summer of 2021. Twenty-one-day-old seedlings were transplanted on June 3 at a density of 21.2 hills m-2. Crop management, such as irrigation, fertilizer applications, pest, and disease control, followed Japan's conventional rice farming systems [2]. The number of transplants per hill was arbitrarily varied from three to five seedlings to facilitate the evaluation of the effectiveness of the method. Images were acquired using an unmanned aerial vehicle (UAV), and the ground truth of the number of tillers on each hill was manually acquired (Figure 2). The UAV flew automatically and photographed the field with the onboard RGB camera. The waypoint of the flight was set up to fly perpendicular to the rows. right side of figure 2 demonstrates the flight. Ground truth tiller numbers were observed for about 250 hills on each date in a 7m x 6m field plot. 

(please look at the attachment(word) to see image)

Figure 2. Experimental field.

2.Selection of patch image is elaborated in the following descriptions.

Considering the lens distortion and occlusion effect due to inadequate elevation angle,  we only observed about 2m2 (1m × 2m) area in the center of a single image. Furthermore, seedlings that fluttered heavily in the wind were removed from the analysis. In the latter growth stage, rice plants tend to be affected by wind heavier, and more images were removed from the analysis. Finally,  ten out of thirty patches were extracted and selected from each image.

3.Statistical method and replication, software tools were elaborated in the following descriptions. 

A computer with a 64-bit Windows 10 operating system (Core i5 1.2 GHz, 8 GB RAM, Surface) was used to process and analyze the image data. The algorithm was written in Python using Google Colaboratory (Google Corporation, Menlo Park, CA, USA). The images acquired were saved in a JPG file format. The flowchart of image processing is shown in (Figure 4). These steps will be described further in the following sections. The following libraries were used for the thinning process. (https://github.com/magikerwin1993/Line-Following-Python) The following libraries were used for the decorrelation stretching. (https://github.com/lbrabec/decorrstretch) The rest of the program is self-coded. After ground truth tiller number and estimated leaf tip number were obtained, statistical analysis such as correlation coefficient analysis and T-tests were conducted using Excel 2019 (Microsoft Corporation, Redmond, WA, USA).

In result section, statistical analysis is elaborated. Underlined description is added.

The statistics of the observation results are described below (Table 2). For correlation coefficients, ground truth tiller number and manually counted leaf-tip number were 0.7–0.8 regardless of the growth stage; manual and automatically counted leaf-tip number were 0.5–0.3, and ground truth tiller number and automatic leaf-tip counting was 0.3–0.2. T-test between (A) (B) (C) was also conducted and the P value was less than 0.01 percent in most cases. True tiller number (A), Manually counted leaf-tip number (B), Automatically counted leaf-tip number (C) has statistically significant differences.

L191: Elaborate the Results portion. Write in continuous.  …………..observation of the manually measured value. Judging from the average of multiple…………………….

Response : Line break at “value. Judging…” was deleted, and result figure was elaborated after that. 

The results of image estimation of the number of tillers per hill were compared with those of field observation of the manually measured value. Judging from the average of multiple hills, the standard deviation becomes smaller (Figure 7). For 25 hills average, the error variance dropped to about 1.0, and the error average dropped to 1.1 on July 3 and August 7. The error average remained above 3 on July 23.

 L194: Why samples were different size? 155, 71 and 90? Why not same for three date?

Response : In the latter growth stage or windy conditions, rice plants tend to be affected by wind heavier, and more images were removed from the analysis(Please look at the figure below). The following description is added in material & method chapter.

Considering the lens distortion and occlusion effect due to inadequate elevation angle,  we only observed about 2m2 (1m × 2m) area in the center of a single image. Furthermore, seedlings that fluttered heavily in the wind were removed from the analysis. In the latter growth stage, rice plants tend to be affected by wind heavier, and more images were removed from the analysis. Finally,  ten out of thirty patches were extracted and selected from each image.

(please look at the attachment word file to see image)

L210: Yes, please mention statistical method and softwere used in materials and methods section.

Response : The following description is added to the paper.

A computer with a 64-bit Windows 10 operating system (Core i5 1.2 GHz, 8 GB RAM, Surface) was used to process and analyze the image data. The algorithm was written in Python using Google Colaboratory (Google Corporation, Menlo Park, CA, USA). The images acquired were saved in a JPG file format. The flowchart of image processing is shown in (Figure 4). These steps will be described further in the following sections. The following libraries were used for the thinning process. (https://github.com/magikerwin1993/Line-Following-Python) The following libraries were used for the decorrelation stretching. (https://github.com/lbrabec/decorrstretch) The rest of the program is self-coded. After ground truth tiller number and estimated leaf tip number were obtained, statistical analysis such as correlation coefficient analysis and T-tests were conducted using Excel 2019 (Microsoft Corporation, Redmond, WA, USA).

L214: Increase Fig. 8 size. If possible, make the axis in black color for more visibility. Same for Fig. 9 and 10 and further.

Response : Black axis were added and figure size were increased as below.

(please look at the attachment word file to see image)

L218: Instade of this (B) Manually counted leaf-tip number – (C)Automatically counted leaf-tip number Write comparison between Manually counted leaf-tip number(B)  and Automatically counted leaf-tip number(C).

L204: Denote (A) (B) (C) and (D) at the end. Like True tiller number (A)

Response: The text has been revised to state the sentence first through manuscript.

L299: increase font size in Fig. 11

Response : Font size is increased in the figure as below.

(please look at the attachment word file to see image)

L329: Improve Conclusion. Add data brief data about the experiments. Tiller count and error data.

Response : The following description was added in conclusion.

 A nondestructive method for observing the tiller number of rice plants for this stage did not exist, and this method showed moderate accuracy. The estimated tiller number in a 1.1 m × 1.1 m area is significantly correlated with the actual number of tillers (error average = 1.02 of error standard deviation = 1.09 at the early reproductive stage), with 60% of hills having an error less than ±3 tillers.

L349: Kindly check references style with journal guideline.

Response : I apologize for the inconvenience,  the style follows the guideline below.

Author 1, A.B.; Author 2, C.D. Title of the article. Abbreviated Journal Name Year, Volume, page range.

We appreciate your review.

Round 2

Reviewer 1 Report

One major question to be addressed is the motivation behind the chosen pixel sizes (page 9 and Figure 6).

There are some grammatical errors in the updated contexts that need to be fixed. 

Author Response

We appreciate your comments
